# Experimental and Numerical Studies on Hot Compressive Deformation Behavior of a Cu–Ni–Sn–Mn–Zn Alloy

**DOI:** 10.3390/ma16041445

**Published:** 2023-02-09

**Authors:** Yufang Zhang, Zhu Xiao, Xiangpeng Meng, Lairong Xiao, Yongjun Pei, Xueping Gan

**Affiliations:** 1State Key Laboratory for Powder Metallurgy, Central South University, Changsha 410083, China; zhangyufun@126.com (Y.Z.); meng@bowayalloy.com (X.M.); 2School of Materials Science and Engineering, Central South University, Changsha 410083, China; xiaolr@csu.edu.cn; 3Key Laboratory of Non-Ferrous Metal Materials Science and Engineering, Ministry of Education, Changsha 410083, China; 4Ningbo Boway Alloy Material Co., Ltd., Ningbo 315135, China; yongjun.pei@bowayalloy.com

**Keywords:** Cu–9Ni–6Sn, hot deformation, processing map, microstucture, finite element analysis

## Abstract

Cu–9Ni–6Sn alloys have received widespread attention due to their good mechanical properties and resistance to stress relaxation in the electronic and electrical industries. The hot compression deformation behaviors of the Cu–9Ni–6Sn–0.3Mn–0.2Zn alloy were investigated using the Gleeble-3500 thermal simulator at a temperature range of 700–900 °C and a strain rate range of 0.001–1 s^−1^. The microstructural evolution of the Cu–9Ni–6Sn alloy during hot compression was studied by means of an optical microscope and a scanning electron microscope. The constitutive equation of hot compression of the alloy was constructed by peak flow stress, and the corresponding 3D hot processing maps were plotted. The results showed that the peak flow stress decreased with the increase in the compression temperature and the decrease in the strain rate. The hot deformation activation energy was calculated as 243.67 kJ/mol by the Arrhenius equation, and the optimum deformation parameters for the alloy were 740–760 °C and 840–900 °C with a strain rate of 0.001~0.01 s^−1^. According to Deform-3D finite element simulation results, the distribution of the equivalent strain field in the hot deformation samples was inhomogeneous. The alloy was more sensitive to the deformation rate than to the temperature. The simulation results can provide a guideline for the optimization of the microstructure and hot deformation parameters of the Cu–9Ni–6Sn–0.3Mn–0.2Zn alloy.

## 1. Introduction

Ultra-high-strength elastic copper alloys have been widely used in the electronic and electrical industries [1]. Beryllium copper is the most widely used elastic copper alloy [2]; however, it has a poor stress relaxation resistance performance at high temperatures. Cu–Ni–Sn alloys have received widespread attention due to their excellent overall mechanical properties and excellent resistance to stress relaxation [3]. The typical Cu–Ni–Sn alloys, including Cu–15Ni–8Sn [4], Cu–9Ni–6Sn [5], Cu–20Ni–5Sn [6], etc., have excellent overall mechanical properties, with hardnesse of 300~380 HV, electrical conductivities of 5~15% IACS, tensile strengths of 800~1100 MPa, and elongations of 3.8~5%. Among them, the Cu–9Ni–6Sn alloy (wt.%, designed as C72700 in an ASTM B740-84 standard) has a good balance of conductivity (of >12% IACS) and comprehensive mechanical properties (a hardness of >300 HV, a tensile strength of >800 MPa, and an elongation of >5%) [7].

Due to differences in the melting points and surface tensions of the alloying elements in the Cu–Ni–Sn alloy, segregation of the alloying elements is easy to occur during casting, which directly affects the subsequent processing performance of the materials. The micro-segregation and some of the structural defects of the alloy ingot could be improved by the hot deformation process, and a suitable hot working process should be necessary for the Cu–9Ni–6Sn alloys to obtain good comprehensive performance [8]. The physical simulation test equipment for hot compression (the “Gleeble” series physical simulation test machine) is used to study the hot deformation behavior of the tested samples, and the hot working conditions of the alloy can be predicted according to the constitutive equation and the hot working diagram established in the dynamic material model (DDM) of hot compression simulations [9]. The deformation behavior and structural evolution of the alloys in the hot compression experiments provide a theoretical basis for the hot deformation process, thus, shortening the development period of the alloys [10]. This method has been successfully applied to various alloys, such as Ni [11], Ti [12], Al [13], Mg [14,15], and Cu [16]. In recent years, studies on the hot deformation behavior of Cu–Ni–Sn alloys have also been reported. Jiang et al. designed a Cu–20Ni–5Sn–0.25Zn–0.22Mn alloy, and a typical thermal processing map of the alloy was obtained. The hot deformation activation energy of the alloy was calculated to be 295.1 KJ/mol, and the optimum processing parameters were determined to be 760~880 °C/0.1~0.001 s^−1^ [17]. Zhao et al. performed hot deformation experiments on Cu–15Ni–8Sn alloys at deformation temperatures between 825 °C and 925 °C and strain rates of 0.0001–0.6 s^−1^. It was found that Si and Ti promoted the dynamic recrystallization nucleation of the Cu–15Ni–8Sn alloys, and Ni_16_Si_7_Ti_6_ particles precipitated, inhibiting the growth of recrystallized grains [18]. Niu compared three models of deformation (the Arrhenius model, the Johnson–Cook model, and the Zerilli–Armstrong model) and found that the Arrhenius model had the highest accuracy with a calculated hot deformation activation energy of 310 kJ/mol for the Cu–15Ni–8Sn alloy [19]. According to the current study, the hot deformation activation energy of copper alloys with different contents of alloying elements varies greatly, with a few obvious rules to follow. Therefore, it is necessary to study the hot deformation behavior of Cu–9Ni–6Sn alloys. Numerical simulation technology is an effective method to optimize the conditions during hot deformation [20]. At the same time, it can help researchers comprehensively understand the microstructure evolution of an alloy during deformation processing [21]. In numerical simulation methods, finite element (FE) is considered an important method to predict thermal mechanical processing. The Deform-2D software, which was used by Jia et al., simulated the hot compression and recrystallization behavior of a Ni alloy, and the comparison of the simulation results with the actual manufacturing results verified the accuracy of the model.

So far, there have been a few reports on the microstructure control and optimization of heat deformation conditions for Cu–9Ni–6Sn alloys by the combination of finite element simulations and hot compression simulation experiments. In this paper, hot compression experiments were performed on the Cu–9Ni–6Sn–0.3Mn–0.2Zn alloy at a strain rate range of 0.001~1 s^−1^ and a temperature range of 700~900 °C. Constitutive equations and thermal processing diagrams of the Cu–9Ni–6Sn alloy were established to fill the gap in the field of high temperature hot deformation of this system alloy. The microstructure evolution of typical regions during the compression process was studied by means of optical microscopy (OM) and scanning electron microscopy (EBSD). The thermal compression process of the alloy was simulated using the Deform-3D finite element software, and the accuracy of the model was verified by comparing the simulation and experimental results.

## 2. Materials and Methods

### 2.1. Materials

The raw materials used in this experiment were obtained by heating them to pure copper, pure nickel, pure tin, pure manganese, and pure zinc in an atmospheric smelting furnace. The composition of the alloy was measured by an inductively coupled plasma optical emission spectrometer (ICP-OES), as shown in Table 1. Figure 1a shows the dendritic microstructure of the casted ingot (as shown in Figure 1a). After being homogenized at 920 °C for 2 h, the dendritic microstructure disappeared and equiaxed grains with an average diameter of about 264.1 μm formed (Figure 1b). Figure 1c shows the XRD results of the homogenized alloy, where the phase composition can be determined [22]. According to the comparison with the Cu PDF card (#99-034), the sample mainly has a diffraction peak of pure copper, where the (200) peak has the largest strength.

### 2.2. Hot Compression

The homogenized alloys were machined into cylindrical specimens with a size of φ6 × 9 mm (a sample size ratio of 1:1.5) according to [23]. The cylindrical specimens were then subjected to isothermal thermal compression experiments on a Gleeble-3500 thermal simulator. According to the common experimental conditions of typical elastic copper alloys for hot deformation experiments [17,19], the testing temperatures were selected as 700 °C, 750 °C, 800 °C, 850 °C, and 900 °C, and the strain rates were selected as 0.001 s^−1^, 0.01 s^−1^, 0.1 s^−1^, and 1 s^−1^, respectively. The alloys were heated to the target temperatures at a heating rate of 5 °C/s before the compression and then held for 3 min to ensure a uniform temperature distribution. Figure 2 shows a schematic diagram of the technological process of the whole experiment. The hot compressed specimens were sliced along the central axis and mechanically or electrolytically polished. Subsequent processing of the sample in the central deformation region is generally performed; thus, samples from this part were selected for microstructure characterization. The microstructure and texture of the hot compressed samples were observed by metallographic microscopy (OM) and the JEOL 7200F field-emission scanning electron microscope (SEM) equipped with the dedicated software. The results were processed and analyzed using the HKL Channel 5 software.

### 2.3. Finite Element Simulation and Verification

#### 2.3.1. Related Condition Settings

The 3D geometric model of the experiment was drawn by the SolidWorks software. As shown in Figure 2, the upper and lower indenters had a size of 110 mm × 10 mm × 1 mm, and the compressed specimens had a size of φ6 × 9 mm. The finite element simulation of the thermal compression process of the cylindrical specimens under different deformation conditions was carried out by the Deform-3D software. The mesh division method was chosen in the form of a tetrahedral cell mesh, and the compressed workpiece was divided into 32,000 mesh cells. The hot compression of the Cu–Ni–Sn alloy is a typical metal plastic forming process; thus, the effect of the elastic deformation on the deformation process was ignored. The simulation type was set to a rigid plastic finite element model with the LaGrange incremental simulation mode. The convergence conditions were as follows: 108 steps, and each step was set to move down by 0.05 mm· s^−1^ step. The speed of each step was 0.0054 mm·s^−1^, 0.054 mm·s^−1^, 0.54 mm·s^−1^, and 5.4 mm·s^−1^ (corresponding to the strain rates of 0.001 s^−1^, 0.01 s^−1^, 0.1 s^−1^, and 1 s^−1^, respectively).

#### 2.3.2. Theoretical Basis of Finite Element Numerical Simulation

The rigid visco-plastic finite element method was used in this paper to simulate and analyze the Cu–9Ni–6Sn alloy in the hot compression process, which is applicable to the plastic deformation process of thermally processed and strain rate sensitive materials [24]. The rigid visco-plastic finite element method was based on the following assumptions [24]:

(1) Ignore the effect of elastic deformation and bulk force of the material;

(2) Material deformation obeys the Levy–Mises flow theory;

(3) The material is homogeneous and incompressible, and its volume remains constant throughout the deformation process;

(4) The loading condition gives the boundary between the rigid and plastic regions;

When plastic deformation of the rigid visco-plastic materials occurs, the plastic mechanics can be expressed by the following basic equations:

(1) Balanced equations:
(1)
$$\sigma_{ij,j}=0$$

where *σ_ij_* is the Cauchy stress component in the workpiece;

(2) Velocity–strain rate relationship equation:
(2)
$${\dot{\varepsilon }}_{ij}=\frac{1}{2}\left(u_{i,j}+u_{j,i}\right)$$


(3) Levy–Mises stress–strain rate relationship equation:
(3)
$${\dot{\varepsilon }}_{ij}=\dot{\lambda }{\sigma_{ij}}^{\prime }$$


(4)
$$\dot{\lambda }=\frac{3}{2}\frac{\dot{\bar{\varepsilon }}}{\bar{\sigma }}$$

where 
$\dot{\bar{\varepsilon }}=\sqrt{\frac{2}{3}{\dot{\varepsilon }}_{ij}{\dot{\varepsilon }}_{ij}}$
 is the rate of equivalent effect strain, and 
$\bar{\sigma }=\sqrt{\frac{3}{2}{\sigma_{ij}}^{\prime }{\sigma_{ij}}^{\prime }}$
 is the equivalent effect stress;

(4) Mises yield guidelines:
(5)
$$\frac{1}{2}{\sigma_{ij}}^{\prime }{\sigma_{ij}}^{\prime }=k^{2}$$

where 
$k=\frac{\bar{\sigma }}{\sqrt{3}}$
 is for ideal rigid plastic materials, and *k* is the constant.

(5) Volumetric incompressible conditions:
(6)
$${\dot{\varepsilon }}_{v}={\dot{\varepsilon }}_{ij}\delta_{ij}=0$$


(6) Boundary conditions:

The boundary conditions were divided into force surface boundary conditions and velocity boundary conditions, respectively:
(7)
$$\mathrm{on}\;\mathrm{the}\;\mathrm{stress}\;S_{F},\;\sigma_{ij}n_{ij}=F_{i}$$


(8)
$$\mathrm{on}\;\mathrm{the}\;\mathrm{speed}\;S_{U},\;{\dot{u}}_{i}={\bar{u}}_{i}$$


(7) Constitutive relationship (the Arrhenius model) [25]:
(9)
$$\dot{\varepsilon }=A{\left[\sinh\left(\alpha \sigma \right)\right]}^{n}\exp\left(Q/RT\right)$$

where 
$\dot{\varepsilon }$
 is the strain rate, σ is the ture stress, *Q* is the heat of the deformation activation energy, *R* is the ideal gas constant 8.314 J/(mol·K), *T* is the hot deformation temperature, and *A*, *α*, and *n* are the constants, respectively.

### 2.4. Hot Deformation Intrinsic Model

In recent years, scholars have built on empirical observations and proposed to describe the dependence of alloy rheological stress on the strain rate, strain, and temperature with respect to stress by building an intrinsic model, e.g., the Johnson–Cook model [26,27], the Khan–Liang–Farrokh model [28], the Molinari–Ravichandran model [29], the Voce–Kocks model [30], and the Arrhenius model. Among them, the Arrhenius model and its improved model are relatively widely used. Sellars and McTegart first proposed the Arrhenius model in 1966 [31]. The Arrhenius model is expressed as follows:
(10)
$$\dot{\varepsilon }=A{\left(\sin\alpha \sigma \right)}^{n}\exp\left(-Q/RT\right)$$


In order to broaden the temperature range for the application of the Arrhenius model, the Zener–Hollomon parameter was used to further modify the model [25,32], which is as follows:
(11)
$$Z=A{\left[\sin\left(\alpha \sigma \right)\right]}^{n}=\dot{\varepsilon }\exp\left(Q/RT\right)$$


## 3. Results & Discussion

### 3.1. Effect of Deformation Conditions on the True Stress–Strain Curve

Figure 3 shows the true stress–strain curves of the Cu–9Ni–6Sn alloy, which is hot compressed at different temperatures and strain rates. In the initial stage of strain, the flow stress increased significantly with the increasing strain. The value of stress fluctuated in a small range after reaching the peak stress. It was found that the flow stress curves of 0.001 s^−1^ and 1 s^−1^ at 700 °C continued to increase with the increasing strain. The typical dynamic recovery (DRV) behavior was observed at 750 °C and 0.1 s^−1^ (Figure 3c), while the typical dynamic recrystallization (DRX) behavior was observed at 900 °C and 1 s^−1^ (Figure 3d). It showed a sawtooth shape in the middle part of the true stress–strain curve under the condition of 800 °C/0.001 s^−1^, which could be associated with the incidence of dynamic recrystallization [33,34].

Figure 4 indicates the peak stresses of the samples at different conditions. The peak stress increased with the increasing strain rate and decreased with the increasing deformation temperature. When the strain rate was 1 s^−1^, the peak stress decreased from 263 MPa at 700 °C to 115 MPa (at 900 °C). As the strain rate increased from 0.001 s^−1^ to 1 s^−1^, the peak stress of the samples deformed at 700 °C increased from 82 MPa to 263 MPa.

In the initial stage of deformation, a large number of dislocations appeared in the matrix. The entanglement between the dislocations or the interaction between the dislocations and grain boundaries lead to a rapid increase in the strength of the alloy; thus, the flow stress of the alloy increased sharply. After reaching the peak of flow stress, the deformation of the alloy was completed in a short time at the high strain rate. Therefore, the dislocation density and flow stress values in the matrix were also quite high. At the low strain rate, the alloy underwent significant reduction and recrystallization because the hot deformation temperature was higher than the recrystallization temperature [35].

### 3.2. Flow Stress Constitutive Equation

The Arrhenius kinetic equation, which is summarized by Sellars et al., describes the thermal processing properties of a material through the characteristic parameters in the true stress–strain curve. The relationship between these characteristic parameters, such as the peak flow stress, strain rate, and temperature of processing, could be described as a unified constitutive equation [26] as follows:
(12)
$$\dot{\varepsilon }=A{\left[\sinh\left(\alpha \sigma_{p}\right)\right]}^{n}\exp\left(-\frac{Q}{RT}\right)={\left[\sinh\left(\alpha \sigma_{p}\right)\right]}^{n}\exp\left(\mathrm{InA}-\frac{Q}{RT}\right)$$

where 
$\sigma_{p}$
 is the peak stress.

Under low stress conditions (or 
ασ<0.8
) and high stress conditions (or 
ασ>1.2
), the constitutive equation can be written using Equations (13) and (14), respectively, which are as follows:
(13)
$$\dot{\varepsilon }=A_{1}\sigma_{p}^{n_{1}}\exp\left(-\frac{Q}{RT}\right)$$


(14)
$$\dot{\varepsilon }=A_{2}\exp\left(b\sigma_{p}\right)\exp\left(-\frac{Q}{RT}\right)$$

where *A_1_*, *A_2_*, *n*_1_, and *b* are the constants, and *α = b/n*_1_.

Taking logarithms for each side of Equations (1)–(3):
(15)
$$\ln\dot{\varepsilon }=\ln A+n\ln\left[\sinh\left(\alpha \sigma_{p}\right)\right]-\frac{Q}{RT}$$


(16)
$$\ln\dot{\varepsilon }=\ln A_{1}+n\ln\sigma_{p}-\frac{Q}{RT}$$


(17)
$$\ln\dot{\varepsilon }=\ln A_{1}+b\sigma_{p}-\frac{Q}{RT}$$


The linear relationship of 
$\ln\dot{\varepsilon }\sim \ln\left[\sinh\left(\alpha \sigma_{p}\right)\right],\;\ln\dot{\varepsilon }\sim \ln\sigma_{p},\;\ln\dot{\varepsilon }\sim \sigma_{p}$
 was obtained. The three relationships were plotted separately, as shown in Figure 5. As the thermal compression temperature increased, the slopes of the fitted curves all showed an increasing trend, except for the slope of 
$\ln\dot{\varepsilon }\sim \ln\sigma_{p}$
 the fitted curve at 800 °C, which showed a sudden decrease. By linear fitting, it can be calculated that b and *n*_1_ are equal to 0.0534 and 3.36, respectively.

Under the same temperature during the thermal processing, the hot deformation energy reflects the sensitivity of the deformation resistance to the strain rate, and it also directly represents the ease of deformation. For a given temperature and strain rate, the thermal activation energy *Q*, defined in [32] is as follows:
(18)
$$Q=R\left\{\frac{\left(\partial \ln\varepsilon {}^{\circ}\right)}{\partial \ln\left[\sinh\left(\alpha \sigma_{p}\right)\right]}\right\}\left\{\frac{\partial \ln\left[\sinh\left(\alpha \sigma_{p}\right)\right]}{\partial \left(\frac{1}{T}\right)}\right\}=RNS$$

where *R* is the ideal gas constant, *N* is the average value of the slope obtained by fitting 
$\ln\dot{\varepsilon }\sim \ln\left[\sinh\left(\alpha \sigma_{p}\right)\right]$
 at a given temperature, and *S* is the average value of the slope obtained by fitting 
$\ln\left[\sinh\left(\alpha \sigma_{p}\right)\right]\sim 1000/T$
 at a given strain rate (as shown in Figure 6a).

The obtained hot deformation energy of the alloy (*Q*) was 243.67 kJmol^−1^. The relationship between the strain rate and the processing temperature during thermal processing can be determined by temperature-compensated strain rate parameters, which Zener–Hollomon described as follows:
(19)
$$Z=\varepsilon \exp\left(\frac{Q}{RT}\right)=A{\left[\sinh\left(\alpha \sigma_{p}\right)\right]}^{n2}$$


Taking the logarithm of both sides:
(20)
$$\ln Z=\ln A+n2\ln\left[\sinh\left(\alpha \sigma_{p}\right)\right]$$


Figure 6b shows the linear relationship obtained by fitting the line of 
$\ln Z\sim \ln\left[\sinh\left(\alpha \sigma_{p}\right)\right]$
, where the intercept on the y-axis is In*A*, from which the value of the parameter *A* can be obtained. The value of the slope *n*_2_ (3.36) is averaged with the slope of the linear relationship between 
$\ln\dot{\varepsilon }\sim \ln\left[\sinh\left(\alpha \sigma_{p}\right)\right]$
 to obtain the final value of *n*. The calculated results for each parameter of the constitutive equation are shown in Table 2.

Substituting the averaged parameter values into Equation (14), the constitutive equations of the alloy during hot deformation can be obtained as follows:
(21)
$$\dot{\varepsilon }={\left[\sinh\left(0.0113\sigma \right)\right]}^{3.35}\exp\left(22.93-\frac{243670}{RT}\right)$$


As the parameters of the flow stress curve are calculated by several linear fittings, the flow stress curve obtained from the parameters often has some errors compared with the actual measured curve. In order to evaluate the accuracy of the flow stress instantonal equation, the peak stresses under various test conditions were calculated according to the equation and compared with the experimental values. The relative error between the measured and calculated values was calculated using Equation (22), and the results are shown in Figure 7.

(22)
$$\left(\sigma_{c}-\sigma_{m}\right)/\sigma_{m}\times 100\%$$

where σ*_c_* is the peak stress calculated from Equation (21), and σ*_m_* is the actual measured peak stress. According to the calculation results, the minimum and maximum relative errors were 0.04% and −10.8%, respectively, and the average relative error was 4.4%. It was demonstrated that the calculated Arrhenius equation has a small error range and can accurately predict the variation of peak stress in the Cu–9Ni–6Sn alloy.

### 3.3. Hot Deformation Processing Map

According to the dynamic material modeling (DMM) theory, the hot deformation constitutive equation of the Cu–9Ni–6Sn alloy can be expressed [36] as follows:
(23)
$$\sigma =K{\dot{\varepsilon }}^{m}$$


The strain rate sensitivity coefficient of the alloy was obtained by taking the partial derivative of Equation (23) as follows:
(24)
$$m=\frac{\partial \ln\sigma }{\partial \ln\dot{\varepsilon }}$$

where 
$\dot{\varepsilon }$
 is the strain rate, 
σ
 is the flow stress, *m* is the strain rate sensitivity factor, and *K* is the constant. According to the dissipative structure theory, the power consumed by the alloy can be expressed [37] as follows:
(25)
$$P=\sigma \dot{\varepsilon }=G+J=\int_{0}^{i}\sigma d\dot{\varepsilon }+\int_{0}^{\sigma }\dot{\varepsilon d\sigma }$$

where *G* is the dissipation quantity, and *J* is the dissipation coefficient. The strain rate sensitivity factor *m* is defined as follows:
(26)
$$m=\frac{dJ}{dG}=\frac{\dot{\varepsilon }d\sigma }{\sigma d\dot{\varepsilon }}={\left[\frac{\partial \left(\log\sigma \right)}{\partial \left(\log\dot{\varepsilon }\right)}\right]}_{\tau ,\;\;T}\approx \frac{\Delta \log\sigma }{\Delta \log\dot{\varepsilon }}\;$$


When the strain rate and the hot deformation temperature are constant, the dissipation coefficients of the alloy can be expressed as follows:
(27)
$$J=\int_{0}^{\sigma }\dot{\varepsilon }d\sigma =\frac{m}{m+1}\sigma \dot{\varepsilon }$$


At this time, the system has a steady-state flow stress, and the value of *m* is between 0 and 1, indicating that the dissipation state of the system is nonlinear dissipation. In addition, when the value of *m* is 0, the system does not dissipate energy. When the value of *m* is 1, the system dissipation state is ideal dissipation, and the dissipation coefficient of the alloy is at its maximum value (*J_max_*) at this time, which is as follows:
(28)
$$J_{max}=\frac{\sigma \dot{\varepsilon }}{2}$$


The energy dissipation efficiency factor of the alloy (
η
) at a certain strain rate and hot deformation temperature can be obtained as follows:
(29)
$$\eta =\frac{J}{J_{max}}=\frac{2m}{m+1}$$


Ziegler’s continuous instability condition can be expressed as follows:
(30)
$$\frac{\partial D}{\partial \dot{\varepsilon }}<\frac{D}{\dot{\varepsilon }}\;$$

where *D* is the dissipation function. Under the conditions of the dynamic material model, *D* is equal to the dissipation coefficient *J*.

Therefore, the flow instability criterion of the alloy can be expressed [38] as follows:
(31)
$$\xi \left(\dot{\varepsilon }\right)=\frac{\partial \ln\left(\frac{m}{m+1}\right)}{\partial \ln\dot{\varepsilon }}+m<0$$


Based on the true stress–strain curves obtained from the hot compression deformation experiments of the Cu–9Ni–6Sn alloy, the steady-state flow stress values of the true strain of the alloy can be obtained from 0 to 1 at different strain rates and deformation temperature conditions. The flow values and strain rates at each temperature can be fitted with a cubic polynomial, and the strain rate sensitivity and energy dissipation efficiency coefficients of the alloy can be derived from Equations (26) and (29). The flow instability intervals were obtained from the hot deformation temperature and strain rate, and the flow instability intervals of the alloy were superimposed with an energy dissipation diagram to obtain a thermal processing diagram of the sample under the steady-state flow conditions [39], as shown in Figure 8. The alloy instability region was located in the upper left corner of the thermal processing diagram at 700~820 °C/0.01~1 s^−1^. The optimum deformation parameters of the Cu–9Ni–6Sn alloy should be between 740~760 °C with a strain rate of 0.01~0.001 s^−1^ and 840~900 °C with a strain rate of 0.001 s^−1^.

### 3.4. Microstructure Evolution

The microstructure of the hot compression alloy in the central region of the sample under different compression conditions is shown in Figure 9. The 0.9 strain thermal processing map can be divided into the following three regions: unsafe deformation region “A”, safe deformation region “B”, and suitable processing region “C”. The OM observations of the samples were performed in the three typical regions. The samples in the destabilization region were characterized by obvious microcracks on the grain boundaries. Very few dynamically recrystallized grains developed near the large angular grain boundaries of the original grains under the low temperatures or the high strain rates. Increasing the deformation temperature or decreasing the strain rate (region “B” or “C”) resulted in a significant increase in the number of dynamically recrystallized grains. Increasing the temperature or decreasing the strain rate provided more favorable conditions for atomic diffusion, dislocation sliding, and grain boundary migration in the alloy [36].

### 3.5. EBSD Analysis

In order to further investigate the hot processing structure of the Cu–9Ni–6Sn alloy, the samples treated at 700 °C/1 s^−1^, 800 °C/0.001 s^−1^, and 900 °C/0.001 s^−1^ were selected as the typical unsafe deformation region, the safe deformation region, and the suitable processing region for EBSD analyses, respectively. Figure 10 shows the IPF results of the samples deformed under different conditions. The symbols 1, 2, 3, and 4 represent the IPF image, the grain boundary distribution, the recrystallized grain distribution, and the fraction of the recrystallized grain. The corresponding results of the sample deformed at 700 °C/1 s^−1^ (the unsafe deformation region) are shown in Figure 10a. There were many fine recrystallization grains at the original grain boundaries and inside the grains of the alloy. The distribution of the grain boundary angle shows that the sample was dominated by the small angle grain boundary, and the proportion of the large angle grain boundary was only 11.01%, as shown in Figure 10(a2). It can be seen that the grains of the sample under this condition were mainly deformed ones, accounting for 87.32%, as shown in Figure 10(a4). The boundary of the original grain with a large number of recrystallized grains, as shown in Figure 10(a1), was magnified, with the results shown in Figure 10(b1). The recrystallization grain size was about 1.7 μm. Figure 10(c1) shows the IPF of 800 °C/0.001s (the safe deformation region). The grain size of the sample was about 31.84 μm. Figure 10(c2) shows that the fraction of large angle grain boundaries in the sample is about 5.11%, suggesting that its corresponding small angle grain boundary increases compared with the sample deformed at 700 °C/1 s^−1^. Figure 10(c3,4) shows that the sample is still dominated by the deformed grains, but the proportion of recrystallized grains decreases and the proportion of substructure grains increases. Figure 10(d1) shows the IPF map at 900 °C/0.001 s^−1^ (the suitable processing region). The grain size of the sample was about 84.41 μm. At a high temperature and low strain rate deformation, the recrystallized grains grew. Figure 10(c2) shows that the proportion of the large angle grain boundary of the sample is about 13.14%. Figure 10(c3,4) shows that the sample is dominated by substructure grains.

Figure 11 shows several typical recrystallization textures in the central region of the samples deformed under different compression conditions. The sample deformed at 700 °C/1 s^−1^, as shown in Figure 11(a1), had more than 65% of Copper texture, while the sample deformed at 800 °C/0.001 s had the highest Goss texture content of nearly 45%, followed by Cube, as shown in Figure 11(b1). Table 3 shows several typical FCC metal recrystallization textures and their Miller Indices [40]. The sample deforrmed at 900 °C/0.001 s^−1^ had 32% of Q type, 26% of rotated Cube, and 17% of rotated Copper textures, respectively. Figure 11(a2–c2) shows the local average gradient (LAM) in the central region of the samples deformed under different hot-pressing conditions. Figure 11(a3–c3) shows the distribution of LAM. The highest average local gradient was 1.8 for the 700 °C/1 s^−1^ deformed sample (the unsafe deformation region), while the average local gradient was 0.5 for the 900 °C/0.001 s^−1^ deformed sample (the suitable processing region). The lowest local gradient was found in the optimum processing region, and the average local gradient was proportional to the geometric dislocation density, suggesting that the geometric dislocation density and the stored energy were low in the optimum processing region.

### 3.6. Finite Model Simulation Results and Validation

#### 3.6.1. Simulation of Effective Strain Distribution under Different Deformation

Figure 12 shows the simulated distribution of the effective strain field of the 1/2 cylindrical Cu–Ni–Sn alloy specimen in the hot compression tests under different deformation conditions, where the *x*-axis indicates the effective strain and the *y*-axis indicates the percentage of different equivalent strains in the cross section. It shows the inhomogeneity of the effective strain distribution under different deformation conditions. The effective strains were symmetrically distributed along the radial centerline and axis of the specimens, showing a distribution phenomenon of dispersion from the center to the periphery. In contrast to Figure 12a–d, the maximum effective strain in the center of the specimen profile increased with the increasing strain rate at any given deformation temperature. In comparison to Figure 12e,f, the maximum effective strain also increased with the increasing deformation temperature at any given strain rate. The effective strain field distributions in Figure 12c (Figure 8, region “A”) and Figure 11f (Figure 8, region “C”) in the suitable processing region are very uniform.

#### 3.6.2. Simulation of Effective Strain Distribution under Different True Strain

During the hot compression deformation, the internal structure and stress distribution of the alloy can change at any time. In order to explore the internal distribution rule of the samples in the three typical areas and analyze the equivalent strain changes of the alloy in the process of hot compression deformation, the simulation results at 700 °C/1 s^−1^, 800 °C/0.001 s^−1^, and 900 °C/0.001 s^−1^ deformation conditions were used to analyze the equivalent strain change rule of the Cu–9Ni–6Sn alloy during hot compression deformation. According to Figure 13, the drum shape of the sample is quite obvious, especially at 700 °C/1 s^−1^. The distribution of the equivalent strain in the sample was extremely uneven. The maximum value of the equivalent strain was mainly concentrated at the edges of the upper and lower end surfaces and the core of the sample, while the value of the equivalent strain at the center of the end face of the sample was small. The equivalent strain at the waist of the sample varied greatly, and this area was less constrained; it is known as the free deformation area [40]. During the compression process, the equivalent strain value of the sample gradually increased, with the true strain increasing from 30% to 100% and the maximum equivalent strain increasing from 0.347 to 4.03 for the sample deformed at 700 °C/1 s^−1^ (Figure 11a), from 0.287 to 1.85 for the sample deformed at 800 °C/0.001 s^−1^ (Figure 11b), and from 0.285 to 2.11 for the sample deformed at 900 °C/0.001 s^−1^ (Figure 11c), respectively. The difference in the effect of the equivalent force values between the central region and the edge part of the sample in the unsafe deformation zone “A” (700 °C/1 s^−1^) was the largest throughout the compression.

#### 3.6.3. The Change Pattern of Strain Value in Different Regions

In order to further investigate the trend of the equivalent variation with time in the different locations of the two typical regions, the core large deformation region (P1) and the edge free deformation region (P2) [41] were selected for point tracking analysis (as shown in Figure 14a), and the results are shown in Figure 14b,c. All of the equivalent effect variation values of the different deformation regions increased with time, and the growth rate of the free variation region was lower than that of the core. The difference between P1 and P2 under the condition of 900 °C/0.001 s^−1^ was smaller, indicating that the deformation was more uniform. In contrast, the final equivalent variation of the core (deformed at 700 °C/1 s^−1^) was about 1.2, and the free deformation was about 0.6.

## 4. Conclusions

In this work, the hot compression deformation behaviors of the Cu–9Ni–6Sn–0.3Mn–0.2Zn alloy were investigated, and the microstructure evolution and finite element simulation results during hot deformation were discussed. The main conclusions were as follows:

(1) The flow stress of the Cu–9Ni–6Sn–0.5Mn–0.2Zn alloy during hot compression experiments is sensitive to the process parameters, with the peak stress decreasing with the increasing deformation temperature and the decreasing strain rate.

(2) The thermal activation energy of the Cu–9Ni–6Sn–0.5Mn–0.2Zn alloy was 243.67 kJ mol^−1^, and the hot deformation constitutive equation was determined as follows:
$$\dot{\varepsilon }={\left[\sinh\left(0.0113\sigma \right)\right]}^{3.35}\exp\left(22.93-\frac{243670}{RT}\right)$$


(3) Thermal processing diagrams were plotted for the alloys with the rheological instability zone temperatures ranging from 700 °C to 820 °C and the strain rates ranging from 1 s^−1^ to 0.01 s^−1^. The optimum hot deformation parameters for the alloys were 740~760 °C with strain rates of 0.01~0.001 s^−1^ and 840~900 °C with strain rates of 0.001 s^−1^.

(4) According to the Deform-3D finite element simulation results, the distribution of the equivalent strain field in the hot-pressed specimens were inhomogeneous. The alloy was more sensitive to the processing rate compared to the processing temperature. The simulation results can provide a guideline for optimizing the microstructure and hot deformation parameters of Cu–9Ni–6Sn alloys.

## Figures and Tables

**Figure 1 materials-16-01445-f001:**
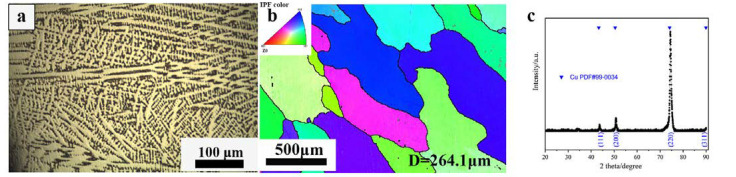
Microstructure of the Cu–9Ni–6Sn alloy before the test: (**a**) optical microstructure of the casted alloy; (**b**) IPF map of the homogenized alloy from an ND direction; (**c**) XRD results of the homogenized alloy.

**Figure 2 materials-16-01445-f002:**
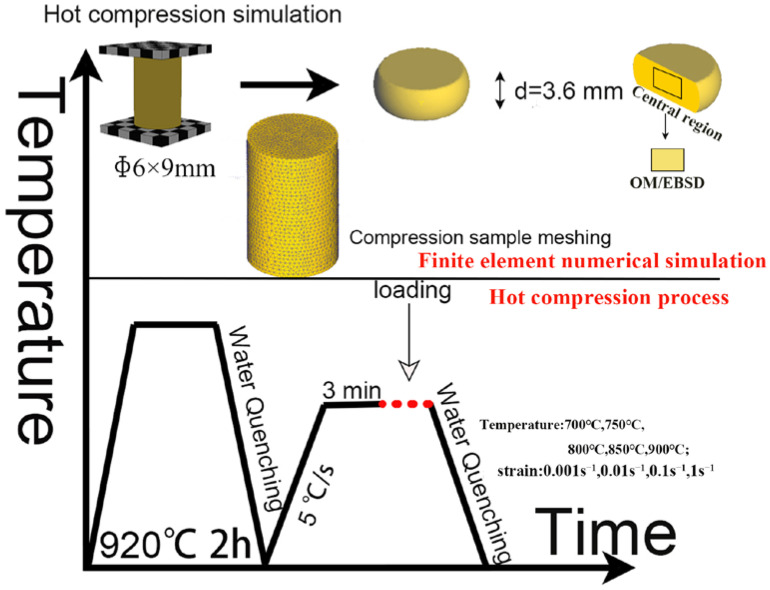
Schematic diagram of the technological process of the whole experiment.

**Figure 3 materials-16-01445-f003:**
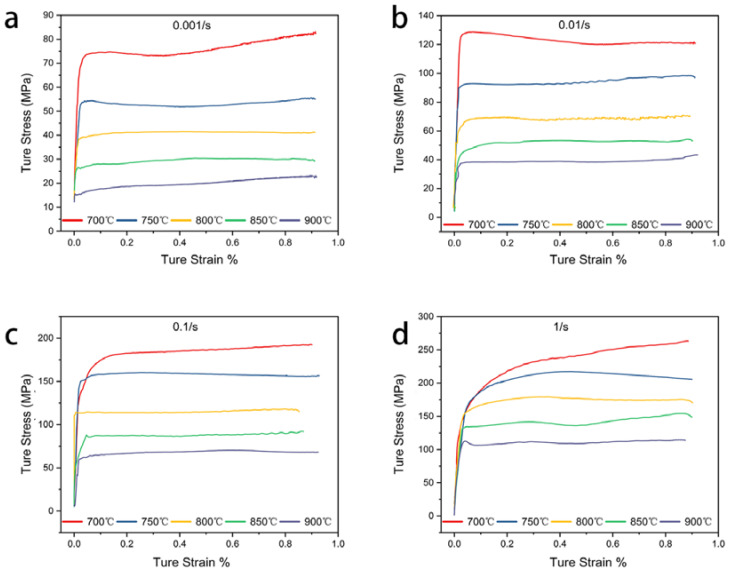
True stress–strain curves of the Cu–9Ni–6Sn alloy at different deformation temperatures and strain rates: (**a**) 0.001 s^−1^; (**b**) 0.01 s^−1^; (**c**) 1 s^−1^; (**d**) 1 s^−1^.

**Figure 4 materials-16-01445-f004:**
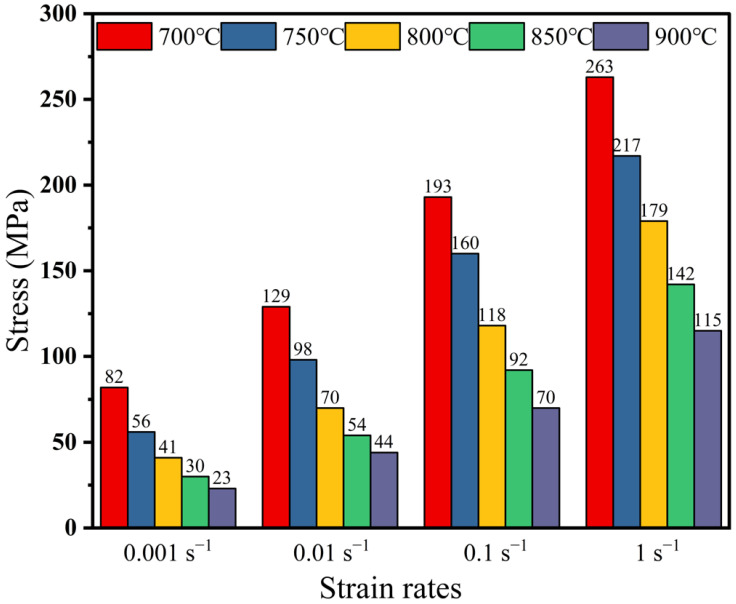
Peak stress of the Cu−9Ni−6Sn alloy under different deformation conditions.

**Figure 5 materials-16-01445-f005:**
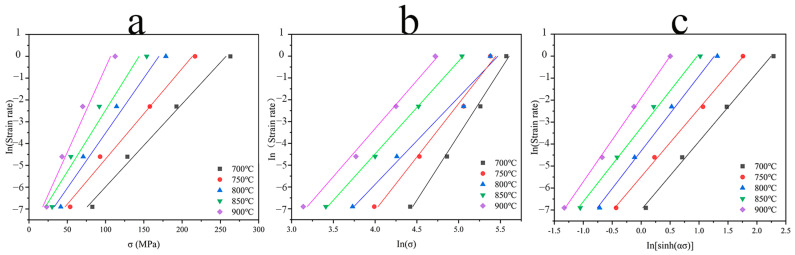
Relationship between peak stress and strain rate: (**a**) 
$\;\ln\dot{\varepsilon }\sim \sigma_{p}$
 curve; (**b**) 
$\ln\dot{\varepsilon }\sim \ln\sigma_{p}$
 curve; (**c**) 
$\;\ln\dot{\varepsilon }\sim \ln[\sinh(\alpha \sigma_{p})]$
 curve.

**Figure 6 materials-16-01445-f006:**
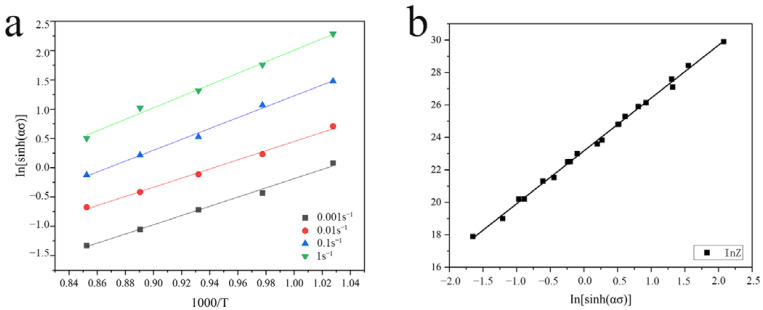
The linear relationship of (**a**) 
$\ln\left[\sinh\left(\alpha \sigma_{p}\right)\right]\sim \ln\frac{1000}{T}$
 and (**b**) 
$\ln Z\sim \ln\left[\sinh\left(\alpha \sigma_{p}\right)\right]$
.

**Figure 7 materials-16-01445-f007:**
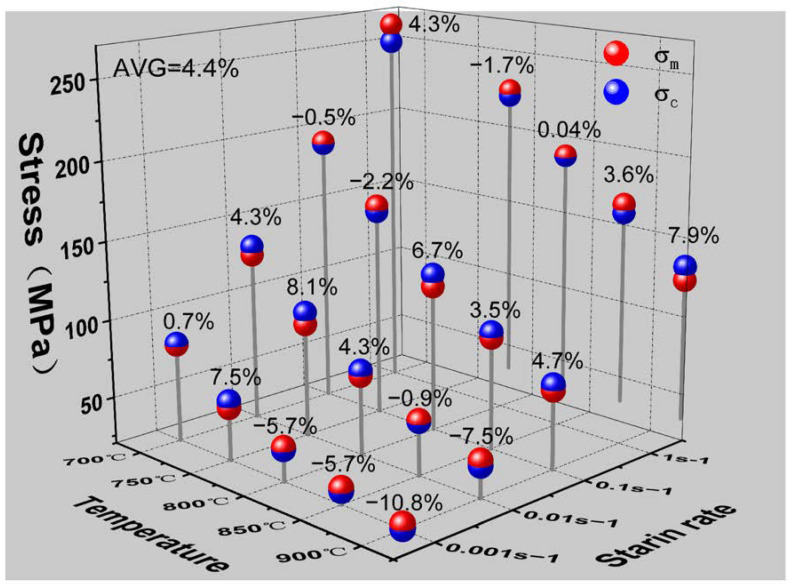
Correlation between the experiment and measured values of the peak stress.

**Figure 8 materials-16-01445-f008:**
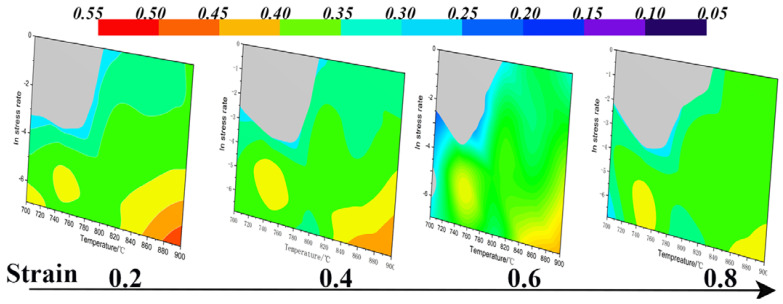
Hot process maps of the Cu−9Ni−6Sn alloy at different true strains.

**Figure 9 materials-16-01445-f009:**
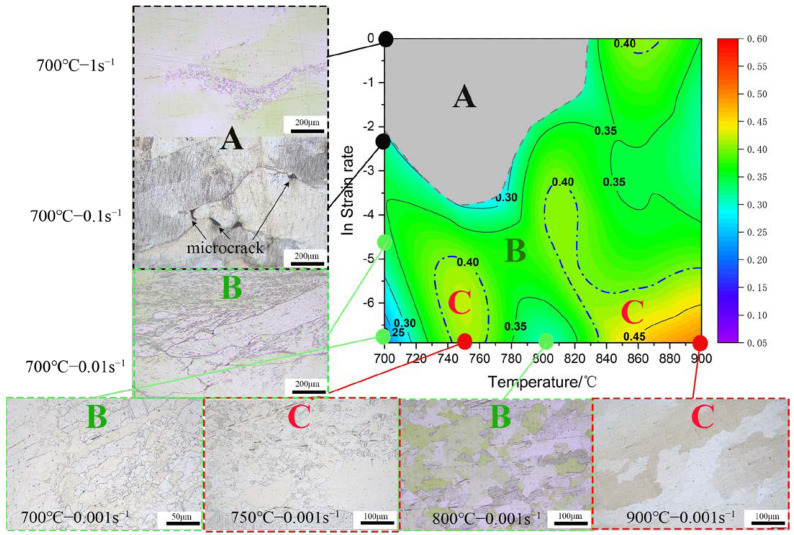
Hot processing map and OM of the Cu−9Ni−6Sn alloy after hot compression with a true strain of 0.9.

**Figure 10 materials-16-01445-f010:**
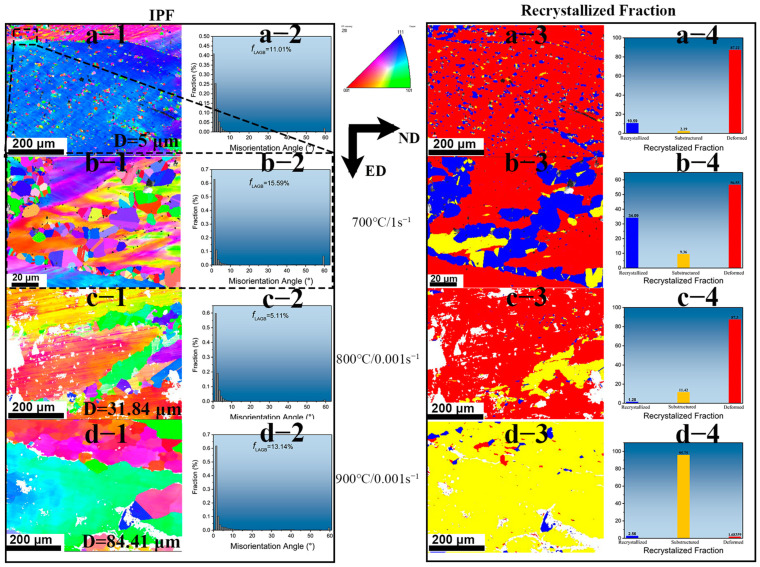
1/2/3/4 are the IPF map/the percentage of grain boundary fraction/recrystallized distribution maps/frequency of recrystallized grain at the center of the sample deformed at: (**a**) 700 °C/1 s^−1^; (**b**) black frame selection from 700 °C/0.001 s^−1^; (**c**) 800 °C/0.001s^−1^; (**d**) 900 °C/0.001s^−1^, respectively.

**Figure 11 materials-16-01445-f011:**
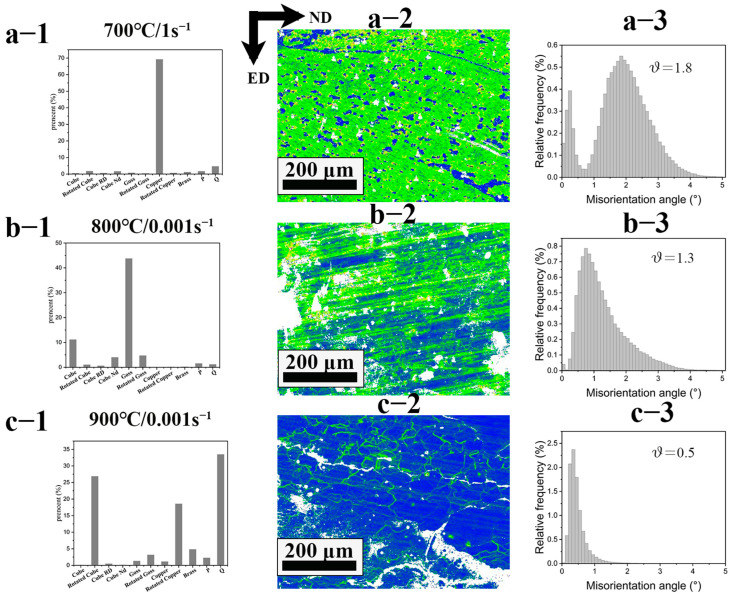
(**a1**–**c1**) show the distribution of several typical recrystallization structures at 700 °C/1 s^−1^, 800 °C/0.001 s^−1^, and 900 °C/0.001 s^−1^, respectively; (**a2**–**c2**) show the local average misorientation (LAM) in the central region at 700 °C/1 s^−1^, 800 °C/0.001 s^−1^, and 900 °C/0.001 s^−1^, respectively; (**a3**–**c3**) are histograms of the distribution of LAM at 700 °C/1 s^−1^, 800 °C/0.001 s^−1^, and 900 °C/0.001 s^−1^, respectively.

**Figure 12 materials-16-01445-f012:**
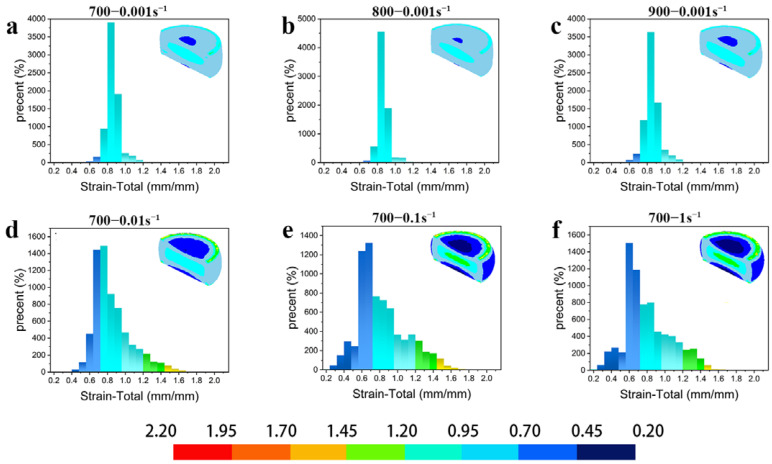
Simulation and prediction of the effective strain (100% of the true strain) distribution on half of the Cu−9Ni−6Sn under different deformation conditions: (**a**) 700−0.001 s^−1^, (**b**) 800−0.001s^−1^, (**c**) 900−0.001s^−1^, (**d**) 700−0.01s^−1^, (**e**) 700−0.1s^−1^, (**f**) 700−1s^−1^, respectively.

**Figure 13 materials-16-01445-f013:**
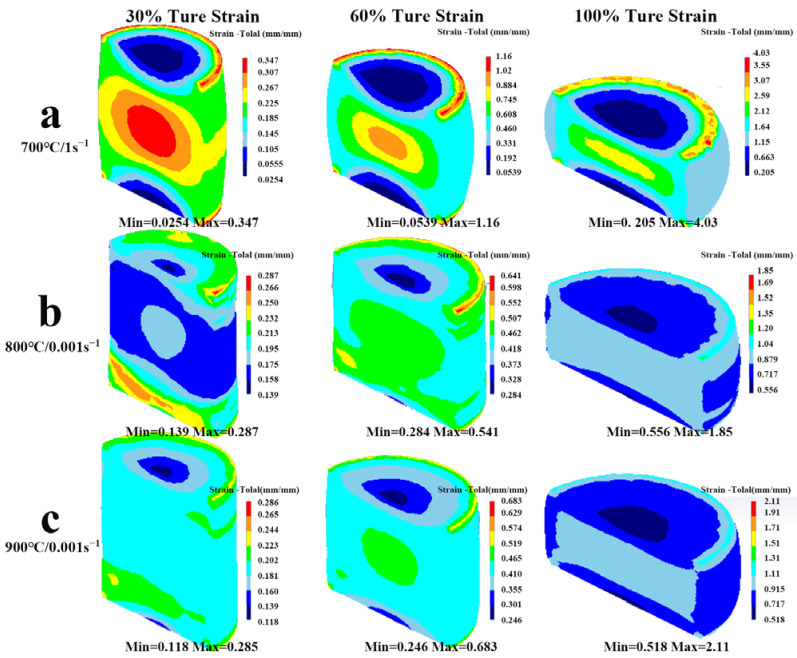
Simulation and prediction of the effective strain distribution of the Cu−9Ni−6Sn alloy under 30%, 60%, and 90% of the true strain at (**a**) 700 °C/1 s^−1^, (**b**) 800 °C/0.001 s^−1^, and (**c**) 900 °C/0.001 s^−1^.

**Figure 14 materials-16-01445-f014:**
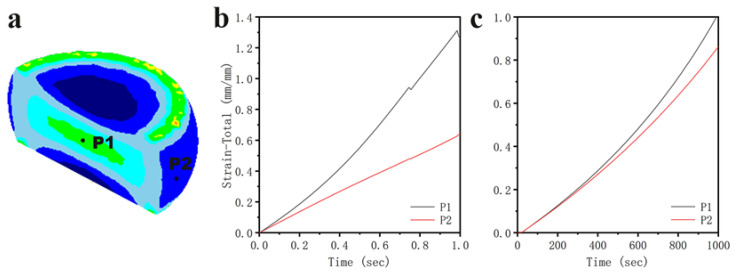
(**a**) Equivalent variation distribution at the end of compression; 700 °C/1 s^−1^ and (**b**) 900 °C/0.001 s^−1^; (**c**) Changes in the strain values at P1 and P2 during compression, respectively.

**Table 1 materials-16-01445-t001:** Composition of Cu–9Ni–6Sn–0.2Mn–0.1Zn (wt.%).

	Ni	Sn	Mn	Zn	Si	P	Cu
Composition	9.06	5.47	0.30	0.18	0.019	0.0016	Bal.

**Table 2 materials-16-01445-t002:** Calculation results of each parameter of the constitutive equation.

Parameter	
b	0.0534
α	0.0113
n	3.35
Q/kJ·mol^−1^	243.67
InA	22.93

**Table 3 materials-16-01445-t003:** Several typical FCC metal recrystallization textures [40].

Designation	Miller Indices {hkl} <uvw>
Cube	{001} <100>
Rotated Cube	{001} <110>
Cube_ND_	{001} <110>
Cube_RD_	{013} <110>
Goss	{011} <100>/{110} <001>
Rotated Goss	{110} <110>
C(Copper)	{112} <111>
Bass	{011} <211>/{110} <112>
Rotated Bass	{110} <111>
P	{011} <122>
Q	{013} <122>

## Data Availability

The data used to support the findings of this study are included within the article.

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
