# Peer review of "Experimental and Numerical Studies on Hot Compressive Deformation Behavior of a Cu–Ni–Sn–Mn–Zn Alloy"

_materials, 2023, doi:10.3390/ma16041445_

Round 1

Reviewer 1 Report

The following, some suggestions for authors:

1.      The abstract should be revised specifying the context of the research, the gap of knowledge, the aim of the work, the materials and the methods adopted and the main results obtained.

2.      The novelty of the work is not clear from Introduction. Which is your main scientific contribution? What is the gap of knowledge that the authors aim to fill? Can you revise this, please?

3.      What do you mean with Composition in table 1? Is it a weight percentage? Is it dimensionless?

4.      Can you describe better Figure 2?

5.      Figure 5 is non-readable, moreover, can you describe the trend of the curves also discussing them?

6.      How did you calculate the parameters reported in Table 2?

7.  All the results obtained were not discussed appropriately within the manuscript.

8.      Can you describe better the experimental validation procedure used for the model?

Careful reading of the text should be done in order to suppress typo errors; can you check them, please?

Reviewer 2 Report

The main problem with this manuscript is that the content is not prepared in standard form, please consider the following points:

1.           The title is not proper. I could suggest "Experimental and numerical study on Hot compressive-deformation behavior of Cu-9Ni-6Sn". This works only contributes to compression behavior, therefore the word "Compression" or "Contraction" must be in the title too.

2.           The abstract should be upgraded in a structured form to have a short sentence of 1) Introduction, 2) Objective, 3) Materials and Method, 4) Result and Discussion, and 5) Conclusion.

3.           The Introduction section is too short and does not contain enough literature review. If this section is not improved substantially, the manuscript could not be considered for publication. You need to provide about two pages and review more than 20 recently published works in the field.

4.           The first paragraph inadequately cites 14 papers. without a proper review of those references.

5.           Please describe another method of metal alloy characterization and review https://doi.org/10.3390/app11073047 and https://doi.org/10.1016/j.jmrt.2021.01.051

6.           The last paragraph of the introduction is not highlighting the gap in science that motivated authors to implement this research.

7.           Section 2: is supposed to be Material and Method, where you systematically explain sample preparation with microstructural images, etc, and complete it with the test program.

8.           Section 2, please provide an image of the sample before the test

9.           how did you obtain the values in table 1? Does any reference need to be cited?

10.         Section 2: You must mention hoe did you decided on the sample geometry and test process. you must cite if you followed any standard method, or take the approach from a published work.

11.         Section 2: you must open a subsection and explain all the necessary information about the simulation as a part of the "Method" to solve the problem. please provide the theoretical part used in the simulation, model and mesh configuration, mesh convergence study, etc.

12.         Section 2 must contain a table of various test and loading condition, that is performed in this work. it is not clear to me now, so I can't judge, because what you have provided in lines 67-69 is different from lines 49-50.

13.         Section results and discussion: suddenly jump into result by providing a  figure with no explanation? Some of the results are provided in the form of a report. For the simulation part, no adequate information is provided in the previous section where you explain how did you create and implemented the FE model.

14.         Figure 3: you have only shown the stress-strain curves, but you must extract the three values of Elastic modulus, yield stress, and ultimate stress from the curves and tabulate them, or show it in a plot form, it is a very valuable result and important to show.

15.         Section 3.1: how did you obtain the stress-strain curves shown in figure 3?  was it a conversion from load-displacement?

16.         Figure 3: case a is true stress and the rest are what stress?

17.         Section 3.2: the main formula and theoretical part are from other published work which should be as a section (theoretical background) before the section on results and discussion. in 3.2, you should only mention your contribution or modification of the formula and result

18.         Section 3.3 should be a subsection of section 3.2

19.         Some axes of Figure 7 are not readable. you must provide a high-quality image of the plot.

20.         Section 3.4 (same as section 3.2): the main formula and theoretical part are from other published work which should be as a section (theoretical background) before the section on result and discussion. in 3.4, you should only mention your contribution or modification of the formula and result.

21.         Image is figure 8 should rotate to the front view only with a clear axis number

22.         Figures 9 and 10, please indicate where did you take the image from? which part of the real sample?

23.        Is the result will be the same if you take from other parts of the sample?

24.         Section 3.7.1; what do you mean by effective strain? which pass did you select to extract the parameter? what is the title of the vertical axis in figure 12? it is not clear

25.         Figure 12: what is the basis to select each row of the plot in figure 12? the bottom row is for 700 and the top for different temperatures? shouldn’t image a and d be similar?

26.         The difference between strain at p1, and p2 of case 700 must be evaluated further, it may be dependent on the applied temperature method which is not been described (you suppose to describe it in the simulation section).

27.         Conclusion: start with no proper description of the manuscript content. This is not acceptable.

Reviewer 3 Report

The submitted article "Hot deformation behavior of Cu-9Ni-6Sn and finite element simulation" has many shortcomings, but in the end it makes a good impression.

Some of the weaknesses are in my comments:

1 The article is not fully formatted in accordance with the journal template

2 Error in last keyword: "element anslysi"

In the Introduction section, authors need to:

3 Write in more detail about the scope of the alloy

4 Make a brief overview of the results of the mentioned references, and not just state them as a fact

5 Write what exactly is the novelty of research

6 Emphasize relevance and clearly state the goal

7 It is necessary to clarify in which plane the structure was viewed

8 Specify whether the temperature heating of the samples during deformation is possible or Gleable instantly compensates for it and the influence of this factor is absent.

9 Has friction been corrected? Why was this particular sample size ratio of 1:1.5 and not 1:2 selected?

10 Make the scales in Figure 3 the same

11 Color about 750 C in figures 3 and 4 should be made the same

12 In section 2.5 Microstructure evolution - should be described in more detail. Almost nothing has been written about the evolution of grains.

13 What was the average size of the elements of the structure?

14 In section 2.6 EBSD Analysis - should be described in more detail.

15 Images must be the same scale.

16 It is necessary to explain what is the reason for the division of the structure into large and small at 700C. Are there any results at 800C?

17 In section 2.6 Finite model simulation results and validation. Degree of deformation e=0.9?

18 It would be interesting to add and analyze the dynamics at different degrees of deformation (for example, e=0.5 and e=0.9)

19 There are a lot of authors in the references only from China, please provide data and international scientists

20 The percentage of self-citation should not exceed 20%

21 References are formatted incorrectly

Round 2

Reviewer 1 Report

The paper was improved accordingly to reviewer's comments.

Author Response

Thank you for your suggestion! 

Reviewer 2 Report

a.       Correct the title to “Experimental and numerical studies

b.       You did some modifications, but the last paragraph of the introduction still requires strong reasoning about the gap of science in the previous study, then the aim/objective of your study to provide the solution to the gap should be provided

c.        The title of section 2 should be “material and method”

d.       In response to my question “8. Section 2, please provide an image of the sample before the test” you have provided “an image: as-cast metallographic diagram”. It is necessary to include it in figure 1

e.       In response to my comment “10. Section 2: You must mention how did you decided on the sample geometry and test process. you must cite if you followed any standard method, or take the approach from a published work”, you have provided two articles in our response. Please cite both of them in the proper place and the manuscript text, mentioning the reasoning for such sample selection.

f.        In response to my comment “12. Section 2 must contain a table of various test and loading condition, that is performed in this work. it is not clear to me now, so I can't judge, because what you have provided in lines 67-69 is different from lines 49-50.”, authors have provided a good response but all the information should be included in the manuscript text.

g.       The conclusion must start with a paragraph describing the manuscript content, then the main findings should be highlighted with bullet points. 

Reviewer 3 Report

The authors did a good job, responded to all my comments and significantly improved the text of the manuscript. The article contains minor typos that can be corrected at the proofreading stage after acceptance. I believe that in its present form the article can be published

Author Response

Thank you for your suggestion! We have made the changes as suggested.